# Diagnosis and Characterization of Plant Viruses Using HTS to Support Virus Management and Tomato Breeding

**DOI:** 10.3390/v16060888

**Published:** 2024-05-31

**Authors:** Enrique González-Pérez, Elizabeth Chiquito-Almanza, Salvador Villalobos-Reyes, Jaime Canul-Ku, José Luis Anaya-López

**Affiliations:** Instituto Nacional de Investigaciones Forestales, Agrícolas y Pecuarias, Celaya, Guanajuato 38110, Mexico; gonzalez.enrique@inifap.gob.mx (E.G.-P.); villalobos.salvador@inifap.gob.mx (S.V.-R.); canul.jaime@inifap.gob.mx (J.C.-K.)

**Keywords:** sRNA-seq, virome, *Solanum lycopersicum*, native germplasm

## Abstract

Viral diseases pose a significant threat to tomato crops (*Solanum lycopersicum* L.), one of the world’s most economically important vegetable crops. The limited genetic diversity of cultivated tomatoes contributes to their high susceptibility to viral infections. To address this challenge, tomato breeding programs must harness the genetic resources found in native populations and wild relatives. Breeding efforts may aim to develop broad-spectrum resistance against the virome. To identify the viruses naturally infecting 19 advanced lines, derived from native tomatoes, high-throughput sequencing (HTS) of small RNAs and confirmation with PCR and RT-PCR were used. Single and mixed infections with tomato mosaic virus (ToMV), tomato golden mosaic virus (ToGMoV), and pepper huasteco yellow vein virus (PHYVV) were detected. The complete consensus genomes of three variants of Mexican ToMV isolates were reconstructed, potentially forming a new ToMV clade with a distinct 3’ UTR. The absence of reported mutations associated with resistance-breaking to ToMV suggests that the *Tm-1*, *Tm-2*, and *Tm-2^2^* genes could theoretically be used to confer resistance. However, the high mutation rates and a 63 nucleotide insertion in the 3’ UTR, as well as amino acid mutations in the ORFs encoding 126 KDa, 183 KDa, and MP of Mexican ToMV isolates, suggest that it is necessary to evaluate the capacity of these variants to overcome *Tm-1*, *Tm-2*, and *Tm-2^2^* resistance genes. This evaluation, along with the characterization of advanced lines using molecular markers linked to these resistant genes, will be addressed in future studies as part of the breeding strategy. This study emphasizes the importance of using HTS for accurate identification and characterization of plant viruses that naturally infect tomato germplasm based on the consensus genome sequences. This study provides crucial insights to select appropriate disease management strategies and resistance genes and guide breeding efforts toward the development of virus-resistant tomato varieties.

## 1. Introduction

Tomato (*Solanum lycopersicum* L.) is one of the world’s most economically valuable vegetable crops and holds cultural and economic importance for Mexico. Viral diseases are among the most damaging threats to tomato crops, causing a significant decrease in yield and quality [1]. The yield reduction induced by viruses can range between 42.1% and 95.5% [2]. However, economic and productive losses vary greatly and often depend on the viral species and the growing region [3].

The management of viral diseases in agriculture relies on two main strategies: immunization to obtain resistant plants and prophylactic measures to restrain virus spread [4]. Among immunization strategies, virus resistance breeding is one of the most promising methods for controlling viral diseases [2]. Identifying both pathogen and host genes influencing disease outcomes is crucial for plant breeding strategies.

Additionally, understanding the inheritance pattern of the plant’s resistance trait, along with the presence of pathogen strains or physiological races, is essential for effective breeding [5]. Therefore, the virus disease management strategy must be developed specifically for each viral species, strain, host, and environment. A successful strategy heavily relies on a fast and accurate identification of the causal agent, as well as knowledge of virus vectors, transmission biology, and ecology [4].

The high number of tomato-infecting viral species and frequent mixed infections with different species or variants of the same species [3,6,7] challenge standard diagnostic tests. Molecular and serological tests require prior knowledge of the target virus and are often unsuitable for detecting unexpected or unknown viruses. Furthermore, their high specificity can limit the detection of diverse isolates, variants, or strains, potentially leading to false negative results [8]. High-throughput sequencing (HTS) offers a powerful solution, identifying plant viruses regardless of their genome nature or structure [9].

In theory, HTS can detect all DNA and RNA viruses present in a sample in a single assay, providing an exhaustive view of a plant’s viral phytosanitary status. HTS can be implemented using individual, bulked, or mixed samples [10,11,12], with reported repeatability and reproducibility of up to 100% [13,14,15]. Moreover, the sequence information obtained can be used for a variety of purposes, including elucidating virus population structure, ecology, and evolution, as well as differentiating variants with varying contributions to disease etiology. Additionally, HTS data can be shared and re-analyzed by multiple users as databases expand [16].

However, major challenges in using HTS for virus detection still arise from the lack of established methodological standards, especially regarding bioinformatic tools, input material, nucleic acid preparation, and the overall cost of the analysis [17].

The sRNA sequencing strategy is used to detect virus-derived small interfering RNAs (siRNA) generated as part of the plant’s gene silencing-based immune response [10]. With this approach, viroids, RNA viruses, DNA viruses, and even persistent viruses can be identified [10,18,19,20,21].

Compared with HTS of total RNA for virus diagnosis, sRNA sequencing yields a very high number of reads and depth of coverage [14,22]. The relative enrichment of viral signatures in sRNA data is likely responsible for the enhanced ability to detect viruses and viroids with relatively low expression levels, which can be challenging using total RNA approaches [23]. Furthermore, the purification of sRNA is simpler compared to with the time-consuming and experimentally challenging methods for other types of viral molecules such as double-stranded RNA or virion purification, which require sophisticated and expensive laboratory equipment such as ultracentrifugation [17,21]. Although the best approach to capture the entire viral diversity is perhaps a combination of all techniques, this strategy is not cost-feasible for routine application in a breeding program.

The cultivated tomato has low genetic diversity due to intensive selection and genetic bottlenecks that emerged during its evolution and domestication [24]. This could explain its high susceptibility to viruses, with over 300 documented viruses, many of which cause disease symptoms and yield losses [3]. To address this, tomato breeding programs require a broader genetic base, readily found in native populations [25,26] and wild relatives [27].

Mexico, a center of tomato genetic diversity, holds native germplasm potentially harboring various degrees of tolerance to biotic and abiotic stresses, along with unique desirable organoleptic characteristics [28]. In a variety of trials conducted in 2021 and 2022 at the Instituto Nacional de Investigaciones Forestales, Agrícolas y Pecuarias, virus-like symptoms were observed in 19 advanced lines derived from native tomatoes collected in different locations of southeastern Mexico. The symptoms, observed over two growing cycles, suggest potential seed-borne viral infection and susceptibility in these advanced lines. Seed-borne viruses pose a particular threat as they can spread widely through seed exchange and establish infected plants within plantations, acting as a continuous source of inoculum [29]. To diagnose the specific viruses, recommend appropriate management strategies, and guide breeding efforts, the virome of these 19 tomato lines was analyzed using High-throughput sequencing of small RNA (sRNA) from leaf tissue composite samples. The analysis was performed in conjunction with PCR and RT-PCR confirmation of the viruses identified in each sample.

## 2. Materials and Methods

### 2.1. Plant Material Sampling

In February 2023, young leaves exhibiting virus-like symptoms were collected from 36 plants of 19 advanced lines of tomato (*Solanum lycopersicum* L.) at a greenhouse evaluation facility at the “Instituto Nacional de Investigaciones Forestales, Agrícolas y Pecuarias” in Celaya, Guanajuato, Mexico (20°34′42.06″ N 100°49′12.37″ W). These advanced lines were developed from native tomatoes collected across six states in Southeast Mexico (Table 1). The observed symptoms included foliar necrosis, local necrotic lesions, yellow mottling, green vein banding, blistering, upward and downward curling, leaf malformation, yellowing, and interveinal chlorosis (Appendix A). 

For high-throughput sequencing (HTS) identification of viruses, foliar tissues were immediately frozen in liquid nitrogen and stored at –72 °C until further processing. To confirm the HTS-identified viruses using PCR or RT-PCR, separate foliar tissues were dried and preserved in silica gel with a moisture indicator, as described by Chiquito-Almanza et al. [30].

### 2.2. Library Construction and sRNA Sequencing

HTS of sRNA and assembly was employed to identify viral species [10]. Total RNA was extracted from approximately 0.4 g of leaf tissue per sample using TRIzol reagent (Invitrogen, Carlsbad, CA, USA) and quantified using a NanoDrop 8000™ analyzer (Thermo Fisher Scientific, Waltham, MA, USA). Agarose gel electrophoresis was used to assess RNA quality. Three composite samples (JM1, JM2, and JM3) were created using pooling equimolar mixtures of total RNA from 15, 10, and 11 plants, respectively (composition details in Table 1). Approximately 3 μg of RNA from each composite sample was shipped to Macrogen Inc. (Seoul, Republic of Korea) for sRNA library construction using the TruSeq^®^ Small RNA Library Preparation Kit (Illumina, San Diego, CA, USA). The libraries were then sequenced on an Illumina HiSeq 2500 (rapid) system in single-end, 50-bp mode.

### 2.3. Small RNA Bioinformatic Analysis

Bioinformatic analysis and virus identification were performed according to Chiquito-Almanza et al. [31], following the procedure outlined by Zheng et al. [32]. Briefly, the quality of raw sRNA reads was assessed using the sRNA_clean.pl script (https://github.com/kentnf/VirusDetect/tree/master/tools/sRNA_clean, accessed on 1 September 2023). This script trimmed 3’ adapter sequences, removed low-quality reads and reads without adapter sequences, and eliminated reads shorter than 15 nucleotides (nt) after trimming. The average number of sRNA mapped and unmapped to the host was obtained by aligning the trimmed sRNA sequences (15–40 nt) to the tomato genome (NCBI: GCA_000499845.1) using the Bowtie v1.1.1 program [33].

De novo assembly of reads into contigs and subsequent virus identification were conducted using the online VirusDetect pipeline with default parameters (coverage = 0.1 and depth = 5; http://virusdetect.feilab.net/cgi-bin/virusdetect/index.cgi, accessed on 8 March 2024). The tomato reference genome and plant virus reference database group 248 U100 available on the VirusDetect platform were used for host sRNA subtraction and guided assemblies through sRNA sequence alignment.

To determine consensus sequences, the longest contigs from each library were mapped to the corresponding virus reference genome for each virus identified. For ToMV, the 3′ end was additionally amplified with RT-PCR and Sanger sequencing. The RT-PCR was performed on each of the 36 samples using the specific direct primer ToMV_GSPD1 (5′-GGTTGCAATTCGGTCTGCTAT-3′) designed from the assembled contigs and the reverse primer ToMV_3R from the study by Lyu et al. [34], with a 16 (T > A) base substitution (3′-TTATATATGGGCCCCTACCG-5′). These primers flank a 124 nt region of the coat protein (CP) end and the complete 3′ untranslated region (UTR) of ToMV. The consensus and the contig sequences of each virus were aligned against their respective taxa using the NCBI database with BLASTn [35]. Consensus viral genomes were reconstructed and annotated using SnapGene V4.1.9. Software (GSL Biotech LLC, Chicago, IL, USA). Protein molecular weights were predicted using the Compute pI/Mw tool (https://web.expasy.org/compute_pi/, accessed on 11 march 2024).

### 2.4. Recombination and Phylogenetic Analysis

The phylogenetic analysis was performed using the complete genome sequences of the isolates of ToMV available in the NCBI GenBank (Appendix A). The MEGA11 software package version 11.0.13 [36] was utilized for nucleotide sequence alignment, selection of the best-fit substitution model, construction of the phylogenetic tree, and mutation rate. Nucleotide sequences were aligned using the ClustalW algorithm with default settings. The number of mutations and mutation rates were determined by removing alignment sites containing gaps. The maximum–likelihood method with 1000 bootstrap replicates was employed for phylogenetic tree construction. A tobacco mosaic virus (TMV) isolate (NC_001367) was used as an outgroup. Nodes with bootstrap values less than 70% were collapsed. Assessment of potential recombination events among ToMV isolates was carried out using the RDP v.4.101 program [37] on the same sequence set used for phylogenetic analysis. A putative recombination event was considered significant if supported by at least four of the seven implemented methods, with an associated *p*-value < 1 × 10^−6^.

### 2.5. PCR and RT-PCR Confirmation of HTS

To confirm the presence of the viruses identified with HTS, total nucleic acids (DNA and RNA) were extracted from each of the 36 leaf tissue samples preserved in silica gel. The extraction method employed the cetyl trimethyl ammonium bromide (CTAB) protocol described by Abarshi et al. [38], with the modifications outlined in Chiquito-Almanza et al. [30]. The integrity of total nucleic acids was assessed using 1% agarose gel electrophoresis. Quality and quantity were determined with spectrophotometry, measuring absorbance at 260 nm and 280 nm wavelengths using a NanoDrop 8000™ instrument (Thermo Fisher Scientific, Waltham, MA, USA). For all samples, reverse transcription (RT) was performed using 1 µg of total nucleic acids and a SuperScript III First-Strand synthesis kit (Invitrogen, Carlsbad, CA, USA) in a 20 µL reaction volume. The RT products were stored at −20 °C and subsequently used as templates for the detection of both DNA and RNA viruses with PCR and RT-PCR, respectively. Specific primers reported by Yan et al. [39] were used for RT-PCR detection of ToMV. Detection of PHYVV and ToGMoV was achieved using PCR with primers described by Nakhla et al. [40].

## 3. Results

### 3.1. HTS Data

Three sRNA libraries, designated JM1Lc, JM2Lc, and JM3Lc, were constructed from total RNA composite samples JM1, JM2, and JM3 (Table 1). These sRNA libraries were subjected to HTS, generating between 72.0 and 72.4 million raw reads per library. Following adapter trimming and removal of low-quality reads, 58.9, 57.4, and 58.5 million clean reads within the 15–40 nucleotide range were obtained from the JM1Lc, JM2Lc, and JM3Lc libraries, respectively. Between 90.6% and 92.7% of these reads mapped to the host tomato genome, while 4.3, 5.4, and 4.8 million reads were identified as non-host.

**Table 1 viruses-16-00888-t001:** sRNA composite libraries from advanced tomato lines of native tomatoes from southeastern Mexico and virus confirmation using PCR/RT-PCR.

Sample/ Library ^a^	Tomato Lines	Fruit Shape	Municipality	State	Samples	PCR or RT-PCR Confirmation		
						ToGMoV	PHYVV	ToMV
JM1/JM1Lc	JCM02	Riñón	Huachinango	Puebla	JCM02-1	+	−	+
					JCM02-7	−	−	+
	JCM05	Cherry	Xoxocotla	Morelos	JCM05-1	−	−	+
					JCM05-2	−	−	+
	JCM10	Riñón	Huachinango	Puebla	JCM10-2	−	−	+
					JCM10-7	+	−	+
	JCM11	Riñón	Zitlala	Puebla	JCM11-2	+	−	+
					JCM11-4	+	−	+
	JCM14	Chino criollo	Altepexi	Puebla	JCM14-1	−	−	+
	JCM15				JCM15-2	−	−	+
	JCM16	Chino criollo	Zinacatepec	Puebla	JCM16-4	−	−	+
					JCM16-6	−	−	+
	JCM17	Chino criollo	Miahuatlán	Puebla	JCM17-3	−	−	+
	JCM18	Chino criollo	Tehuacán	Puebla	JCM18-4	+	−	+
	JCM19	Chino criollo	Zinacatepec	Puebla	JCM19-1	−	−	+
JM2/JM2Lc	JCM03	Riñón	Tlacolula	Oaxaca	JCM03-1	−	−	+
					JCM03-5	−	−	+
					JCM03-11	−	−	+
	JCM04	Riñón	Poza Rica	Veracruz	JCM04-2	−	−	+
					JCM04-9	−	−	+
					JCM04-13	+	−	+
	JCM06	Cherry	Tlacolula	Oaxaca	JCM06-1	+	−	+
	JCM09	Medio saladette	Tlacolula	Oaxaca	JCM09-1	+	−	+
					JCM09-8	+	−	+
	JCM12	Riñón	Zozocolco	Veracruz	JCM12-2	+	−	+
JM3/JM3Lc	JCM01	Riñón	Dzitbalché	Campeche	JCM01-6	−	−	+
					JCM01-7	−	−	+
					JCM01-9	−	−	+
					JCM01-11	−	−	+
	JCM07	Riñón	Teapa	Tabasco	JCM07-1	−	−	+
					JCM07-5	−	+	+
					JCM07-11	−	−	+
	JCM08	Medio riñón	Dzitbalché	Campeche	JCM08-2	+	−	+
					JCM08-3	+	−	+
					JCM08-4	+	−	+
	JCM20	Riñón	Dzitbalché	Campeche	JCM20-2	−	−	+

^a^: Composite sample and library names. +: Detected. −: Not detected.

### 3.2. Virome Characterization of Tomato Advanced Lines

After the VirusDetect process, between 3 and 11 contigs exhibiting nucleotide identity to DNA-A and DNA-B components of ToGMoV were assembled from the three sRNA libraries, and 10 and 2 contigs with nt identity to DNA-A and DNA-B components of PHYVV were assembled from the JM3Lc library. The contigs that showed the identity to begomoviruses nearly covered the complete viral genomes, with 90.9–99.7% coverage and high nucleotide identity of 90.0–98.4% compared with the respective reference viral genome sequences (Table 2). Furthermore, one contig that nearly covered the complete genome reference sequence with >99% nt identity to ToMV was assembled from each library (Table 2). No other viruses were identified.

Consensus sequences exhibiting the typical genome features described for DNA-A and DNA-B components of PHYVV and ToGMoV were generated by mapping the contigs from each library against their respective reference genomes (Appendix A). Three nearly complete consensus sequences of the ToGMoV DNA-A component comprising 2599, 2583, and 2608 nt with 99% nt identity among them were reconstructed, exhibiting the highest nucleotide identity (97.5–98.4%) with the Rioverde SLP2 DNA-A isolate (EF501976.1) from Mexico. Similarly, the 2546 nt PHYVV DNA-A consensus sequence shared 93.3% nt identity with the DNA-A component of the PEL95W2007 isolate (LN848879.1), also originating from Mexico.

The presence of PHYVV, ToGMoV, and ToMV in all the samples was confirmed with PCR and RT-PCR. PHYVV was identified only in the sample JCM07-5 in co-infection with ToMV. ToGMoV was identified in 13 samples in co-infection with ToMV, and ToMV was identified in all samples in single and mixed infection (Table 1).

### 3.3. Analysis of ToMV Reconstructed Sequences

To perform the recombination, mutation rates, and phylogenetic analysis, and identify mutations reported as associated with resistance-breaking, the complete consensus genome sequences of ToMV were determined. For this purpose, the assembled contigs from composite samples JM1, JM2, and JM3 were used as a reference, and the complete 3’ UTR sequence (264 nt) was obtained with RT-PCR and Sanger sequencing using cDNA synthesized from each of the 36 samples (Table 1).

The contigs assembled from composite samples JM1 and JM2 exhibited 100% nucleotide identity, while the contig from sample JM3 showed 99.9% identity to JM1 and JM2. These contigs, ranging from 6290 to 6300 nt in length, spanned the complete 5’ UTR (72–73 nt), four open reading frames (ORFs), as well as a 107–118 nt fragment of the 3’ UTR (Appendix A). No differences were detected across the nucleotide sequences of the 3’ UTR of the 36 samples analyzed. As a result, the full-length consensus genomes, designated INIFAP JM1, INIFAP JM2, and INIFAP JM3, were reconstructed and deposited in the NCBI GenBank database with the accession numbers PP481218–PP481220.

While INIFAP JM1 and INIFAP JM2 have identical 6447 nt sequences, INIFAP JM3 has 6448 nt, with a cytosine insertion at the beginning of the 5’ UTR and silent mutations at positions 3114 (G > A) and 3117 (A > G) compared with INIFAP JM1 and INIFAP JM2. The ORF analysis revealed the presence of four ORFs encoding the putative viral replicase (126.3 kDa), the RNA-dependent RNA polymerase (RdRp) (183.5 kDa), the movement protein (MP) (29.2 kDa), and the CP (17.7 kDa).

The 3’ UTR region of INIFAP JM1–3 differed from most 3’ UTR sequences of ToMV isolates available in the NCBI GenBank database. The alignment of this region with the 3’ UTR of 52 ToMV isolates revealed a 63 nt insertion present solely in the 3’ UTR of INIFAP JM1–3, and the isolate SRVP22_05 (OQ722333).

The highest nucleotide identity of the complete consensus genome of INIFAP JM1–3 ranged from 99.34 to 99.37% with the isolate SRVP22_05 (OQ722333) from Russia. None of the previously reported mutations in ToMV associated with overcoming the resistance conferred by the *Tm-1*, *Tm-2*, and *Tm-2^2^* genes were identified in the reconstructed consensus sequences.

### 3.4. Recombination, Mutation Rates, and Phylogenetic Analysis

The Tamura–Nei model [41] with gamma-distributed rates and invariant sites was determined to be the best-fitting nucleotide substitution model for phylogenetic analysis. Phylogenetic analysis revealed that the INIFAP JM1–3 isolates clustered in a compact group outside the principal clade containing 47 of the ToMV strains from China, Japan, Australia, Germany, Russia, Republic of Korea, and other countries (Clade I). The other eight isolates formed separate branches outside this principal clade, including the Mexican isolates (Figure 1).

Four putative recombination events, supported by at least four of the seven different methods employed, were detected in the ToMV isolates NVWA36783860, Tai’an 2, and N5 from China, and SRVP22_05 from Russia (Table 3). Notably, the third recombination event, identified at positions 6126–6442 nt within the 3’ UTR of the SRVP22_05 isolate, was also present in the isolates 99-1 from the USA, Tianjin from China, and the consensus sequences of Mexican isolates INIFAP JM1–3. Analysis indicated that the major parent for this recombination event was the isolate AH4 from Egypt, while the minor parent remains unknown (Table 3).

The mutation rates of the INIFAP JM1–3 isolates were compared with the reference Queensland isolate from Australia (NC_002692) and the SRVP22_05 isolate. The ORFs of all isolates had the same nucleotide length, but length variations were observed across the 5’ and 3’ UTRs. The mutations in the ORFs range between 2–47 nt and 0.42–1.01 amino acids (aa), with mutation rates between 0–2% and 0–0.76%, respectively (Table 4). In the 5’ UTR, between 0 nt and 2 nt were mutated, with a mutation rate between 0.0% and 2.74%, while in the 3’ UTR, between 11 and 16 nt were mutated, with a mutation rate of 4.8–6.06% (Table 4).

## 4. Discussion

Early and precise detection of viruses, along with cultural practices and immunization, are a crucial component of plant virus disease control strategies [4]. The HTS technology has emerged as a valuable tool that facilitates the identification of phytopathogenic organisms and enhances disease management approaches. HTS enables the detection of novel and emerging pathogens, facilitates the tracking of disease outbreaks, and contributes to the development of disease-resistant cultivars [42,43].

In the current study, sRNA high-throughput sequencing was utilized to identify the viruses naturally infecting advanced lines of native tomatoes. To minimize library construction costs, three composite samples were created by pooling multiple tomato plants and lines (Table 1). Virus identification was performed using the online VirusDetect pipeline and confirmation was realized using PCR and RT-PCR. This strategy was chosen based on our bioinformatic skills, economic capacity, and infrastructure.

We selected VirusDetect software because it is an automated bioinformatic service for plant virus discovery with a public web interface dedicated to the analysis of sRNA data [32]. It is recommended for virologists with low to moderate bioinformatic skills and has proven useful for detecting known viruses and discovering new ones [17,31,44,45,46], including persistent viruses [21,31]. VirusDetect has achieved 100% sensitivity between sequencing depths of 2.5 M and 250 K, without false positives [47]. In our analysis, between 4.3 M and 5.4 M non-host reads were obtained from the libraries comprising 10 to 15 samples. This number of reads corresponds to an average of 286 K to 540 K reads per sample. Therefore, our sequencing depth falls within the highly sensitive range of VirusDetect.

### 4.1. Virus Identification and Implications of Co-Infections

With the implemented strategy, the near-complete consensus nucleotide sequences of the PHYVV, ToGMoV, and ToMV genomes were reconstructed. These consensus sequences represent an overall view of the viruses present in the samples. The confirmation with PCR/RT-PCR (Table 1) not only confirmed the identity of the viruses but also enabled the determination of the presence of single and mixed infections.

Compared with plants infected solely with ToMV, which showed mild mosaic and leaf deformation symptoms, symptoms in plants with mixed viral infections were more severe. Plants co-infected with ToGMoV and ToMV displayed, primarily on basal leaves, yellow mottling, bleaching, localized necrotic lesions, and leaf edge burning. Meanwhile, the plant co-infected with PHYVV and ToMV exhibited leaf rolling and chlorotic local lesions mainly in the apical leaves (Appendix A).

The complex interplay between viruses during viral co-infections can have significant implications for disease development, symptom expression, and virus epidemiology [48]. These viral co-infections can result in diverse interactions between the coexisting viruses, which can lead to at least four different types of interactions: neutralism, synergism, antagonism, and synergism/antagonism [49,50].

Studies on tomato co-infections with other viral species have revealed the complexity of the interactions. For example, the co-infection of the crinivirus *Tomato chlorosis virus* (ToCV) with the begomovirus *Tomato yellow leaf curl virus* (TYLCV) or with the tospovirus *Tomato spotted wilt virus* (TSWV) induced disease synergism. The co-infection of ToCV with TYLCV resulted in increased disease severity, reduced plant growth, and higher viral accumulation [51], while the co-infection of ToCV with TSWV led to rapid death of susceptible cultivars and the breaking of TSWV resistance in resistant plants when pre-infected with ToCV [52]. On the other hand, the co-infection with the begomoviruses *Tomato yellow spot virus* (ToYSV) and *Tomato rugose mosaic virus* (ToRMV) reduced the viral titers of both viruses, but the symptoms were more severe compared with single infections [53].

Understanding the nature of these virus–virus interactions is an important consideration in the management of plant viral diseases, as the outcomes can vary widely depending on the specific species or strain of viruses involved, the environmental conditions, and the cultivar. The implications of the presence of mixed infections, such as the co-infection of ToMV with ToGMoV or with PHYVV, and how this may affect the dynamics and impact of these viral diseases, need to be further investigated.

### 4.2. Genomic Characterization and Phylogenetic Relationships of ToMV Isolates

The integration of the phylogenetic, recombination, and mutation data provides a more thorough picture of the evolutionary dynamics of ToMV, which may have implications for predicting the emergence of new viral variants and designing durable resistance strategies in tomato cultivation.

The reconstructed sequences INIFAP JM1–3 isolates shared the highest nucleotide identity with the sequence of the SRVP22_05 isolate obtained from a wastewater sample [54]. The SRVP22_05 isolate was classified within the order Martellivirales, but no specific species was assigned. BLASTn analysis of its sequence indicated that it shares >99% identity with various ToMV isolates. Its detection in wastewater aligns with the known ability of ToMV to survive in aqueous solutions under different pH and temperature conditions [55,56]. This evidence strongly suggests that SRVP22_05 is a ToMV isolate.

While the SRVP22_05 isolate sequence was reported as a complete genome, the 5’ and 3’ end sequences are likely partial. Alignment of this sequence with 52 complete ToMV genomes showed 23 fewer nucleotides at the beginning of the 5’ UTR, and 35 nt fewer, upstream of the 63 nt insertion, in the 3’ UTR. The SRVP22_05 sequence was obtained through metagenome-assembled genome analysis and not confirmed by Sanger sequencing [54]; therefore, it is possible that the 63 nt insertion in the 3’ UTR is an assembly artifact. However, the sequences of the 3’ UTR of the INIFAP JM1–3 isolates were obtained using Sanger sequencing and were identical across all analyzed samples. The unique 63 nt insertion could be a signature feature of this ToMV lineage and may have implications for its molecular epidemiology and evolution.

Interestingly, Russia is the fourth most important importer of Mexican tomatoes, which could explain the similarity of the isolates in such geographically separated locations. This characteristic could be used for the design of diagnostic assays or to further the understanding of ToMV biology.

A high number of mutations were detected in the ORFs encoding the putative viral replicase (126.3 kDa) and the RdRp (183.5 kDa) of the INIFAP JM1–3 isolates, the majority of which being silent mutations (Table 4). Although none of the previously reported mutations associated with ToMV resistance-breaking [57,58,59,60] were identified, the high rate of mutations identified in the 3’ UTR region (Table 4) suggests a potential genomic divergence and could have important implications for pathogenesis.

The function of the ToMV 3’ UTR is poorly studied, but some evidence suggests this region is important for pathogenesis [61]. In the related tobamovirus TMV, the 3’ UTR contains a complex secondary structure consisting of three consecutive pseudoknots immediately downstream of the CP ORF stop codon, followed by a 3’-terminal transfer RNA (tRNA)-like structure with two additional pseudoknots [62,63]. The upstream pseudoknot domain in TMV has a crucial role in viral replication and translation. Mutational analysis shows this domain could interact with eEF1A/GTP with high affinity [64], and modifications in the TMV 3’ UTR can detrimentally affect infectivity [65,66], virus accumulation, and symptom expression [67]. Given the functional importance of the 3’ UTR in the closely related TMV, the ToMV 3’ UTR likely has similar critical roles.

Furthermore, the detection of recombination events, particularly the one shared between the SRVP22_05 isolate and the Mexican INIFAP JM1–3 isolates, highlights the importance of considering recombination as a mechanism contributing to the genetic variation of ToMV. Interestingly, the third recombination event, with an unknown minor parent (Table 3), coincides with the position of the 63 nt insertion found in the 3’ UTR of the INIFAP JM1–3 sequences. This suggests that recombination may have contributed to the acquisition of this unique genomic feature, which could have implications for the biology or evolution of these Mexican ToMV isolates. Further characterization of the potential functional significance of this 3’ UTR insertion and mutations would be an important area for future research.

The phylogenetic analysis provides insights into the global diversity of ToMV strains. The majority of the strains cluster within a primary clade, which is consistent with the findings reported by Lyu et al. [34]. Notably, the isolates INIFAP JM1–3 form a distinct group. This information in conjunction with the high sequence identity (99.9–100%) between the Mexican isolates suggests a low degree of genetic diversity among these isolates, and a potential regional differentiation of the INIFAP JM1–3 isolates, which could be influenced by geographical isolation, host-pathogen co-evolution, or other factors.

The tomato lines analyzed in this study were developed from native germplasm collected in different states of southeastern Mexico and correspond to accessions with distinct fruit shapes. Due to the ability of ToMV to be seed-transmitted, it is necessary to determine whether the ToMV variants identified originate from any of the collection sites or were acquired at the site where the genetic improvement is being carried out. Further investigations into the evolutionary dynamics and the factors driving the diversification of ToMV in Mexico would be valuable.

### 4.3. Disease Management and Resistance Breeding Implications of the Viruses Identified

Crop breeding efforts may aim to develop resistance against the virome rather than targeting only one or two specific viruses. This approach becomes feasible once the viromes present in different agroecological zones have been thoroughly characterized and when methodologies to expose breeding materials to the relevant viromes during the breeding process [42] and resistance genes are available. The implications for disease management and resistance breeding of the viral species identified in this studio are discussed below.

#### 4.3.1. Begomoviruses ToGMoV and PHYVV

ToGMoV and PHYVV are two circular single-stranded DNA begomoviruses with bipartite genomes comprised of DNA-A and DNA-B components [68,69]. PHYVV is an endemic and widely distributed species in Mexico that infects pepper, *Physalis philadelphica* Lam., and tomato [68,70], while ToGMoV was first reported infecting *S. lycopersicum* and *S. rostratum* in San Luis Potosi State in the north-central region of Mexico in 2007 [69]. Both viruses have also been identified as infecting cucumber plants in Colima, Mexico [71].

Strains of PHYVV compromise the tolerance or resistance of tomato to tomato yellow leaf curl virus (TYLCV), the most damaging and widely distributed monopartite begomovirus [72], and although resistance to mixed infections of PHYVV and pepper golden mosaic virus (PepGMV) has been identified in wild pepper accessions [73], to date, no resistance genes to ToGMoV or PHYVV have been reported in tomato cultivars. Neither of these viruses is seed-transmitted; their transmission occurs in a circulative persistent manner by the vector *Bemisia tabaci* [74]. In general, insect transmission can allow a geographical restriction more than seed transmission, provided that good management practices and strict sanitary measures are implemented to prevent the presence of the vector [75].

#### 4.3.2. ToMV

ToMV is a positive-sense single-stranded RNA tobamovirus [76]. It infects all parts of the tomato plant at any growth stage and can produce a wide range of symptoms such as leaf discoloration and deformation, fruit malformation and size reduction, internal necrotic alterations, and flower abortion [77,78]. This virus is seed-transmitted in susceptible plants and represents a major problem in greenhouse production because it is highly stable, contagious, and cosmopolitan and because it can survive in the soil for several years and remain infectious in nutrient solutions for at least 6 months regardless of temperature and storage conditions [55,56]. Additionally, it can be easily transferred from contaminated workers’ hands, tools, and clothing through simple contact between an infected plant and a healthy one during cultivation practices [79,80]. Therefore, prevention relies on implementing strict sanitary measures through the use of certified pathogen-free propagation material, disinfection of tomato seeds with thermotherapy or chemical treatment [81], sterilizing cutting tools during cultural and manipulation operations, and large crop rotations in soils free of vegetation residues and weeds that can be hosts of the virus [82]. However, genetic resistance is the most suitable option to control this virus.

At present, the three predominant strategies employed for antiviral resistance breeding in tomato crops comprise marker-assisted selection (MAS) for the development of hybrid cultivars, the generation of transgenic lines, and the application of the CRISPR/Cas system for targeted gene editing. Combining conventional hybrid breeding techniques with MAS for introgressing known ToMV resistance genes has proven to be effective, economical, and sustainable. This integrated strategy circumvents current controversies and regulations restricting the use of transgenic crops in many countries [83,84] while expediting the release of improved virus-resistant tomato cultivars.

Three major resistance genes to ToMV have been characterized: *Tm-1*, *Tm-2*, and *Tm-2^2^*. These genes were introgressed into tomato cultivars from the wild tomato species *S. hirsutum* and *S. peruvianum* [78,85]. Among these, *Tm-2^2^* confers resistance to all the most common ToMV strains (0, 1, and 2), except for the 2^2^ strain (not very common in crops) [82,83]; thus, *Tm-2^2^* gene is widely deployed in most breeding programs [86].

Nearly all commercial tomato varieties exhibit high resistance to ToMV, so this viral species is not regarded as one of the economically important viruses affecting tomato crops in Mexico [87]. However, the detection of ToMV variants in all native tomato lines evaluated (Table 1) and the presence of typical symptoms (Appendix A) suggest either of the following: (1) the absence of resistance in the evaluated lines or (2) that these ToMV variants overcome the resistance genes the lines may have. This finding could align with the unimproved genetic background of these materials, as they have not been subjected to breeding efforts targeting ToMV resistance. To harness the genetic diversity present in these native resources, a strategic breeding approach is required for the development of resistant cultivars.

The *Tm-2* and *Tm-2^2^* genes induce a defense reaction in tomato plants by recognizing the movement protein of ToMV [88]. However, this resistance can be overcome by even a few amino acid substitutions in the viral genome. Although no mutations associated with resistance-breaking were identified in the consensus genome sequence of the INIFAP JM1–3 isolates, the high mutation rate, the 63 nt insertion identified in the 3’ UTR region, as well as the amino acid mutations in the ORFs encoding 126 KDa, 183 KDa, and MP, requires determining whether these ToMV variants possess the capacity to overcome the resistance conferred by the *Tm-1*, *Tm-2*, and *Tm-2^2^* genes.

In summary, the implementation of HTS is a preliminary step to the evaluation of the pathogenicity/virulence of the identified viral species. This strategy enables the identification of probable etiological agents to direct management specifically toward the viruses present and their vectors, as well as the identification of viral variants to which the lines are susceptible based on the assembled consensus viral genome sequences. Moreover, this information allows for the selection of resistance genes and the prediction of their effectiveness based on current knowledge of mutations associated with resistance-breaking in these genes.

The evaluation of the capacity of these variants to overcome *Tm-1*, *Tm-2*, and *Tm-2^2^* resistance genes, along with the characterization of advanced lines using molecular markers linked to these resistant genes, will be addressed in future studies as part of the breeding strategy.

## 5. Conclusions

The high-throughput sequencing of sRNA using composite samples, and the confirmation with PCR or RT-PCR of individual samples, allowed us to determine that the advanced tomato lines evaluated were infected with either single or mixed infections of ToGMoV, PHYVV, and ToMV. Furthermore, a potential new clade of ToMV with a distinct 3’ UTR sequence was discovered to be infecting all the evaluated lines. The high mutation rate, the presence of a 63 nt insertion in the 3’ UTR region, as well as the amino acid mutations in the ORFs encoding 126 KDa, 183 KDa, and MP of these Mexican ToMV variants, necessitates evaluating whether the known resistance genes are effective in conferring protection against these variants.

The accurate identification of viral species that naturally infect tomato germplasm using HTS could be part of a strategy for selecting appropriate disease management approaches and guiding resistance breeding efforts.

## Figures and Tables

**Figure 1 viruses-16-00888-f001:**
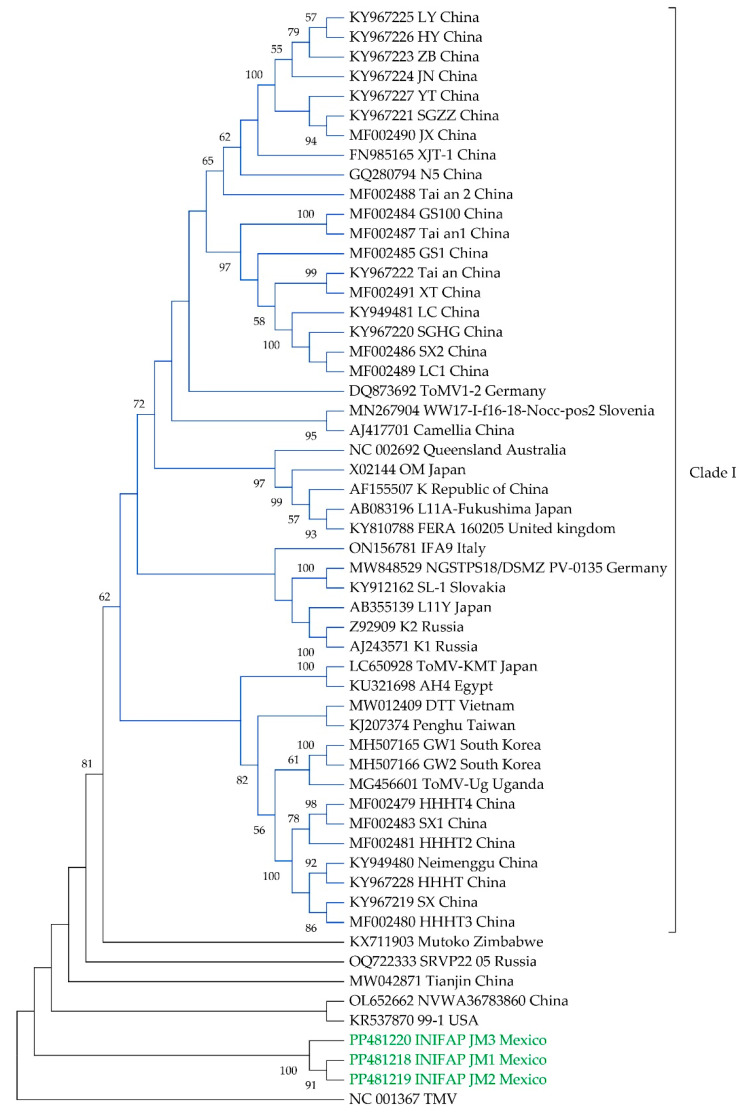
Phylogenetic relationships of tomato mosaic virus Mexican isolates INIFAP JM1–3 with global isolates. The phylogenetic tree was constructed using the complete genome sequences of 55 tomato mosaic virus isolates, including the Mexican isolates INIFAP JM1–3 shown in green color. The tree was inferred using the maximum-likelihood method (TN93 + G + I model) with 1000 bootstrap replicates. A tobacco mosaic virus isolate was included as an outgroup. Nodes with bootstrap support less than 70% are collapsed.

**Table 2 viruses-16-00888-t002:** Identity of viral contigs assembled via high-throughput sequencing of sRNA from composite samples JM1, JM2, and JM3.

Virus Species	Reference Accession ^a^	Genome Length (nt)	Contig Coverage (%)			Number of Contigs			Sequence Identity (%)		
			JM1	JM2	JM3	JM1	JM2	JM3	JM1	JM2	JM3
ToGMoV DNA-A ^b^	EF501976	2615	99.4	98.8	99.7	3	5	3	98.1	97.5	98.4
ToGMoV DNA-B ^b^	DQ406674	2558	90.9	94.5	91.9	7	11	7	90.7	90.2	90.0
PHYVV DNA-A ^b^	LN848879	2630	-	-	96.8	-	-	10	-	-	93.3
PHYVV DNA-B ^b^	LN848915	2595	-	-	99.3	-	-	2	-	-	96.8
ToMV ^c^	KR537870	6383	100	100	99	1	1	1	99.4	99.5	99.3

^a^: GenBank accession number of the viral genome reference sequence used for identification. ^b^: Genus Begomovirus. ^c^: Genus Tobamovirus. -: Not detected.

**Table 3 viruses-16-00888-t003:** Recombination events in complete nucleotide sequences of ToMV isolates detected with RDP4.

Event	Position	Recombinant	Major Parent	Minor Parent	Detection Methods ^a^
1	6214–6442	NVWA36783860 (OL652662)	99-1 (KR537870)	Camellia (AJ417701)	R, G, B, M, C, S, 3S
2	4045–6442	Tai’an 2 (MF002488)	Unknown	GS100 (MF002484)	G, M, C, S, 3S
3 ^b^	6126–6442	SRVP22_05 (OQ722333)	AH4 (KU321698)	Unknown	R, G, B, M, C, 3S
4	3138–5570	N5 (GQ280794)	XJT-1 (FN985165)	(AF155507) ^c^	M, C, S, 3S

^a^: R: RDP, G: GENECONV, B: BootScan, M: MaxChi, C: Chimera, S: SiScan, 3S: 3Seq. ^b^: Event found also in isolates INIFAP JM1 (PP481218), INIFAP JM2 (PP481219), INIFAP JM3 (PP481220), 99-1 (KR537870), and Tianjin (MW042871). ^c^: Attenuated mutant strain.

**Table 4 viruses-16-00888-t004:** Mutation rates of the isolates INIFAP JM1, JM2, and JM3.

Genomic Region (Number of nt/aa)	Isolates	Nucleotide (nt)		Amino Acid (aa)	
		Mutations	Mutation Rate (%)	Mutations	Mutation Rate (%)
5’ UTR (49–73 nt) ^a^	INIFAP JM1 and JM2	1	1.39	-	-
	INIFAP JM3	2	2.74	-	-
	SRVP22_05	0	0.00	-	-
126 KDa (3351 nt/1116 aa)	INIFAP JM1 and JM2	34	1.01	1	0.09
	INIFAP JM3	32	0.95	1	0.09
	SRVP22_05	19	0.57	0	0.00
183 KDa (4851 nt/1616 aa)	INIFAP JM1 and JM2	47	0.97	1	0.06
	INIFAP JM3	45	0.93	1	0.06
	SRVP22_05	23	0.47	0	0.00
MP (795 nt/264 aa)	INIFAP JM1 and JM2	8	1.01	2	0.76
	INIFAP JM3	8	1.01	2	0.76
	SRVP22_05	4	0.50	2	0.76
CP (480 nt/159 aa)	INIFAP JM1 and JM2	3	0.63	0	0.00
	INIFAP JM3	3	0.63	0	0.00
	SRVP22_05	2	0.42	0	0.00
3’ UTR (201–264 nt) ^b^	INIFAP JM1 and JM2	16	6.06	-	-
	INIFAP JM3	16	6.06	-	-
	SRVP22_05	11	4.80	-	-

The number of mutations and mutation rates were determined by removing alignment sites containing gaps. ^a^: The 5’ UTR nucleotide lengths of the Queensland, SRVP22_05, INIFAP JM1, JM2, and JM3 isolates are 71, 49, 72, 72, and 73 nucleotides, respectively. ^b^: The 3’ UTR nucleotide lengths of the Queensland, SRVP22_05, INIFAP JM1, JM2, and JM3 isolates are 201, 229, 264, 264, and 264 nucleotides, respectively. GenBank accession numbers: Queensland (NC_002692); INIFAP JM1 (PP481218); INIFAP JM2 (PP481219); INIFAP JM3 (PP481220); SRVP22_05 (OQ722333). -: Not detected.

## Data Availability

Genomic sequences of viruses described in this study are available at the GenBank database under accession numbers PP481218–PP481220.

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
