# Peer review of "Diagnosis and Characterization of Plant Viruses Using HTS to Support Virus Management and Tomato Breeding"

_viruses, 2024, doi:10.3390/v16060888_

Round 1

Reviewer 1 Report

Comments and Suggestions for Authors

The manuscript addresses an interesting and up-to-date scientific topic as the HTS analysis of advaced lines of tomato, for virus diagnostics and characterization purposes. Sequencing data are used to detect viruses in samples and to obtain full genomes, which are in turn analyzed for mutations and phylogenetic position. 

The manuscript describes the work with appropriate techniques, fitting figures and tables and is properly written. The strategy of pooling samples and confirmation by RT-PCR on single samples is well applied, and the results can realistically be used for the implementation of appropriate phytosanitary measures and for guiding breeding strategies. I suggest publication with minor modifications which are highlighted in the attached revision as comments.

Author Response

REVIEWER 1

Comments and suggestions of reviewer 1 were maded directly on the manuscript.

 Reviewer 1: About the title of the paper the following suggestions were made:

  • “I suggest to rephrase the title as "Diagnosis and characterization" because the work is not only focused on diagnostics but also on the obtainment of complete genomes and their analysis to detect mutations and likely variants, which is the real added value of the manuscript and is used to support management and breeding”
  • “this term is a bit old and belongs to the first years of this technology. I suggest to rather use the current term High Throughput Sequencing (HTS) which is also correctly used throughout the manuscript”

Answer: Both suggestions were accepted and included. The title was modified to: “Diagnosis and Characterization of Plant Viruses Using HTS to Support Virus Management and Tomato Breeding”

Reviewer 1: “I suggest the use of the term "confirmation" rather than "validation". In fact, in plant pathology, validation of a test means the assessment of some performance criteria of the test (i.e., analytical sensitivity, specificity, repeatability etc.), while here you perform the analysis with a different method to ascertain the same result, which is best explained by the term "confirmation". This is valid for all "validation" terms used throughout the manuscript.”

Answer: The term “validation” was changed to “confirmation” throughout the manuscript as rightly suggested by the reviewer.

Reviewer 1: “I suggest to milden this sentence a lot. Although insect transmission is not as rapid and prompt as seed transmission, there are many insect-transmitted viruses which are widespread all over the world (e.g. TYLCV, taking into account begomoviruses). I suggest to rephrase as follows: "In general, insect transmission can allow a geographical restriction more than seed transmission, provided that implementing good management practices and strict sanitary measures are applied to prevent the presence of the vector"

Answer: The suggestion was accepted and included in lines 458-460 as follows: “In general, insect transmission can allow a geographical restriction more than seed transmission, implementing provided that good management practices and strict sanitary measures are applied to prevent the presence of the vector [76].”

Reviewer 1: “I would use the word "accurate" rather than "precise" here”

Answer: The term “precise” was changed to “accurate” throughout the manuscript as suggested by the reviewer.

Reviewer 2 Report

Comments and Suggestions for Authors

This paper reveals the actual infection of the virus in advanced line tomatoes in Mexico. In addition, the authors detected ToMV, which has an unusual sequence in the 3' UTR region, but the overall paper needs to be brushed up.

First, ToMV is an easily transmitted virus. Have you been able to reproduce ToMV infection or disease symptoms by inoculating from one infected plant to another? You should be able to show that it is infectious, not just only sequence information.

You mention the relation between ToMV and resistance breakthrough, but please clarify in the paper which of the resistance genes such as Tm-1, Tm-2, Tm-22, the advanced line possesses in the first place.

Moreover, authors focus on mutations in the 3'UTR and state that it is involved in resistance breaking, but there are also substitutions in the ORF region that involve amino acid mutations. Is there any possibility of a breakthrough involving an unknown amino acid mutation rather than a breakthrough involving a known amino acid?

Also, please state whether the disease was intensified by mixed infection with begomoviruses.

The discussion is redundant and most sentences should be written in the "Introduction" or are unnecessary in the first place. The discussion should contain content that can only be discussed after the results. For example, in chapter 4.1, all except the second paragraph can be written without the results of this study.

Minor points:

The virus names should be written according to ICTV guideline. That is, when referring to the virus itself, it should not be italicized and the head should also be lowercased.

L189: What are the criteria for excepting PHYVV DNA-A and ToGMoV DNA-B?

L195: And no other "known" viruses were found?

L257: "57 Tomato mosaic virus" should be "55 ToMV"

L502: This sentence is important imformation, should move to "Result".

Comments on the Quality of English Language

Abbreviations should be presented the first time and ensure that the abbreviation is used the second and subsequent times. (eg. L76 "RT-PCR" is first time in this MS, but written "reverse transcription (RT)" in L161. L81, "Solanum" is second time in this MS, should be "S.".

Table 2: "pb" in Genome length should be "nt".

L259: Correct "Tobacco" to "tobacco".

Author Response

REVIEWER 2

Reviewer 2: “This paper reveals the actual infection of the virus in advanced line tomatoes in Mexico. In addition, the authors detected ToMV, which has an unusual sequence in the 3' UTR region, but the overall paper needs to be brushed up”

Answer: The article is about the diagnosis of viral infections and the characterization of viruses in advanced tomato lines derived from native tomatoes using HTS. This takes advantage of HTS's capacity to accurately detect all viral species present in a sample by a single test and to obtain the sequences of their genomes. These sequences allow for the identification of virus variants, analyze them for mutations, and perform phylogenetic analysis. Together, this information will allow us to guide management and breeding efforts.

 Reviewer 2: “First, ToMV is an easily transmitted virus. Have you been able to reproduce ToMV infection or disease symptoms by inoculating from one infected plant to another? You should be able to show that it is infectious, not just only sequence information”

Answer: The implementation of HTS is a preliminary step to the evaluation of the pathogenicity/virulence of the identified viral species. This strategy enables identifying probable etiological agents to direct management specifically to the viruses present and their vectors, as well as identifying viral variants to which the lines are susceptible based on the assembled consensus viral genome sequences. Furthermore, this information also allows the selection of resistance genes and the prediction of their utility based on current knowledge of mutations associated with the breaking of resistance in available genes.

Of course, this strategy does not replace Koch's postulates, and in this particular case, the evaluation of the biological characteristics of the identified ToMV variants is still needed. However, these tests are out of the scope of this paper.

Reviewer 2: “You mention the relation between ToMV and resistance breakthrough, but please clarify in the paper which of the resistance genes such as Tm-1, Tm-2, Tm-2, the advanced line possesses in the first place”

Answer: The identification of ToMV variants in the advanced lines, and the presence of typical symptoms, suggest either: 1) the absence of resistance in the evaluated lines, or 2) that these ToMV variants overcome the resistance genes possessed by the lines.

The confirmation of infectivity and the capacity of these identified ToMV variants to break the resistance conferred by Tm-1, Tm-2, and Tm-22, as discussed in lines 504-506. This confirmation and the molecular characterization of the advanced lines are not the subject of this article and will be addressed in future studies.

The sentence in lines 492-495 was rephrased to improve clarity as follows: “However, the detection of ToMV variants in all evaluated native tomato lines (Table 1), and the presence of typical symptoms, suggest either: 1) the absence of resistance in the evaluated lines, or 2) that these ToMV variants overcome the resistance genes the lines may have.”

Reviewer 2: “Moreover, authors focus on mutations in the 3' UTR and state that it is involved in resistance breaking, but there are also substitutions in the ORF region that involve amino acid mutations. Is there any possibility of a breakthrough involving an unknown amino acid mutation rather than a breakthrough involving a known amino acid?”

Answer: We do not state that the identified mutations in the 3’ UTR of the ToMV variants are involved in overcoming resistance in the evaluated lines, as we do not know if the lines possess any resistance genes.

A higher number of mutations were found in the 3’ UTR, as well as in some ORFs, with mostly silent mutations. Even though none of the mutations identified corresponds to those previously reported as related to overcoming the resistance conferred by Tm-1, Tm-2, and Tm-22, other mutations not yet described may be involved in the resistance-breaking of these genes. However, testing this hypothesis is beyond the scope of the present study.

In lines 395-410, we discuss the implications of mutations in 3’ UTR based on current knowledge of the closely related virus TMV, and in lines 500-506, we discuss the need to evaluate whether the novel ToMV variants can overcome the resistance conferred by Tm-1, Tm-2, and Tm-22.

Based on the reviewer's observation, we have improved the wording in lines 504-507 as follows: “…, the high mutation rate, the 63 nt insertions identified in the 3' UTR region, as well as the amino acid mutations in the ORFs encoding 126 KDa, 183 KDa, and MP, requires determining whether these ToMV variants possess the capacity to overcome the resistance conferred by the Tm-1, Tm-2, and Tm-22 genes.”

Reviewer 2: “Also, please state whether the disease was intensified by mixed infection with begomoviruses”

Answer: It is important to consider that the study was realized with plant samples from a normal greenhouse production trial with advanced lines developed from native tomatoes collected in six states in Southeast Mexico. Therefore, to rule out the influence of other factors such as the genetic background of the lines, the presence of other no viral pathogens, or nutritional deficiencies in the development of observed symptoms, further trials are needed, as stated in lines 362-367.

In attention to the right request of the reviewer, we include the following information regarding the symptoms (Lines 345-350): “In comparison with the plants infected solely with ToMV, which showed mild mosaic and leaf deformation symptoms, the disease symptoms in plants with mixed viral infection were more severe. Plants co-infected with ToGMoV and ToMV displayed yellow mottling, bleaching, localized necrotic lesions, and leaf edge burning, primarily on basal leaves. Meanwhile, the plant co-infected with PHYVV and ToMV exhibited leaf rolling and chlorotic local lesions mainly in the apical leaves (supplementary Figure S1).

Reviewer 2: The discussion is redundant and most sentences should be written in the "Introduction" or are unnecessary in the first place. The discussion should contain content that can only be discussed after the results. For example, in chapter 4.1, all except the second paragraph can be written without the results of this study.

Answer: Assuredly, some of the information in chapter 4.1 could be included in the “Introduction”. Since the methodology used in virus detection by HTS is crucial for accurate identification, we decided to include this information as part of the discussion to inform readers not familiar with the use of HTS or with limited bioinformatic skills, like us, the justification, and the theoretic bases of the implemented methodology for HTS analysis.

The observation is appropriate, strictly this section of the discussion will be more applicable if we evaluate different methodologies for library construction and bioinformatic analysis. However, since we consider that the information is useful for some readers, the paragraphs in Lines 322-327, and 334-347 were relocated to the introduction.

Minor points:

Reviewer 2: “The virus names should be written according to ICTV guideline. That is, when referring to the virus itself, it should not be italicized and the head should also be lower cased.”

Answer: The mistake was corrected.

Reviewer 2: “L189: What are the criteria for excepting PHYVV DNA-A and ToGMoV DNA-B?”

Answer: No component was excepted. The sentence was rewritten as follows to improve clarity: “The contigs showing identity to begomoviruses nearly covered the complete viral genomes, with 90.9-99.7% coverage and high nucleotide identity of 90.0-98.4% when compared to the respective reference viral genome sequences (Table 2).

Reviewer 2: “L195: And no other "known" viruses were found?”

Answer: The sentence “No unknown viruses were identified” was replaced by: “No other viruses were identified”

Reviewer 2: “L257: "57 Tomato mosaic virus" should be "55 ToMV"

Answer: The mistake was corrected.

Reviewer 2: “L502: This sentence is important imformation, should move to "Result"

Answer: The information in this sentence was retained in the discussion because the result is previously presented in lines 240-242 as follows: “None of the previously reported mutations in ToMV associated with overcoming the resistance conferred by the Tm-1, Tm-2, and Tm-22 genes were identified in the reconstructed consensus sequences.”

Comments on the Quality of English Language

Reviewer 2: “Abbreviations should be presented the first time and ensure that the abbreviation is used the second and subsequent times. (eg.L76 "RT-PCR" is first time in this MS, but written "reverse transcription (RT)" in L161.

Answer: The abbreviation “RT” was eliminated in L161.

Reviewer 2: “L81, "Solanum" is second time in this MS, should be "S.".”

Answer: The mistake was corrected.

Reviewer 2: “Table 2: "pb" in Genome length should be "nt".

Answer: The mistake was corrected.

Reviewer 2: “L259: Correct "Tobacco" to "tobacco".”

Answer: The mistake was corrected.

Round 2

Reviewer 2 Report

Comments and Suggestions for Authors

In considering the relationship between ToMV and tomato, it is very important to know whether tomatoes have resistance genes such as Tm-1, Tm-2, and Tm-22. Is it possible to determine from the sRNA sequences data whether the advanced tomato lines have these resistance genes (expression)? If so, we can eliminate the possibility of either 1) or 2) below, and we can write the "Discussion" clearly.

"1) the absence of resistance in the evaluated lines, or 

2) that these ToMV variants overcome the resistance genes the lines may have"

If they do not have resistance genes, it is difficult to discuss the relationship between the resistance genes and the results of this experiment in the first place, but if they do, other researchers can treat them as sequence data of resistance-breaking isolates.

L17: Correct "Tomato mosaic virus (ToMV), Tomato golden mosaic virus (ToGMoV), and Pepper huasteco yellow vein virus" to "tomato mosaic virus (ToMV), tomato golden mosaic virus (ToGMoV), and pepper huasteco yellow vein virus" (Do not italicize)

Comments on the Quality of English Language

L16: "RT-PCR" should be "reverse transcriptional PCR"

L98: "RT-PCR" should be "reverse transcriptional (RT)-PCR"
